# Physiologic and molecular response of *Fusarium oxysporum* f. sp. *zingiberi* to ginger autotoxins

**Yan Zhang**[1], **Hongshen Guo**[2], **Yanping Xu**[1], **Xiaochuan Chen**[1], **Miaomiao Zhang**[1], **Naicheng Li**[1]*, **Abdullah Gera**[1]

**1** College of Modern Agriculture and Environment, Weifang Institute of Technology, Weifang, Shandong, China, **2** Anqiu Agrotechnical Extension Center, Anqiu County Bureau of Agriculture and Rural Areas, Weifang, Shandong, China

* nc.li@wfit.edu.cn

## Abstract

Long-term monoculture of ginger triggers *Fusarium* wilt, a disease caused by *Fusarium oxysporum* f. sp. *zingiberi* (*Foz*). However, the interaction between autotoxins and pathogens remains poorly understood. This study examined the allelopathic effects of four autotoxins, syringic acid, coumarin, ferulic acid, and 7-hydroxycoumarin, on the growth, reproduction, and virulence traits of *Foz*. These compounds were previously identified as inhibitors of ginger growth and enzyme activity. The results revealed that the responses of *Foz* varied, likely because of the structural differences among the autotoxins. Syringic acid significantly inhibited mycelial growth, sporulation, and spore germination, while markedly enhancing the activity of cell wall-degrading enzymes and mycotoxin synthesis, as evidenced by the upregulation of *FUB3* and *FUB9*. Coumarin demonstrated a pronounced inhibitory effect on biomass production while concurrently stimulating sporulation and mycotoxin synthesis, as indicated by the upregulation of *FUB6* and *FUB9*. Ferulic acid treatment reduced the activity of cell wall-degrading enzymes and spore germination, while upregulating sporulation and mycotoxin synthesis. A similar pattern was observed with 7-hydroxycoumarin, which exerted a strong inhibitory effect on mycelial growth, biomass production, mycotoxin synthesis, and *FUB* gene cluster expression, except for *FUB1* and *FUB3* expression. In conclusion, four autotoxins exhibited diverse defensive roles against *Fusarium* wilt, whereas the pathogen enhanced its pathogenicity to counter host regulations. This study provides the first evidence of an interaction between ginger autotoxins and *Foz*.

## 1. Introduction

Ginger (*Zingiber officinale* Roscoe) is an herbaceous plant that is generally cultivated as an annual crop despite its inherent perennial nature [1]. This ancient cultivar has been used for centuries in Asian cultures as a culinary spice and for therapeutic

**Data availability statement:** All relevant data are within the manuscript and its Supporting information files.

**Funding:** This work was supported by grants from the Shandong Provincial Natural Science Foundation (grant number: ZR2023QC326) and Key R&D Program of Shandong Province, China (grant number: 2023TZXD028). Li received the first project and Xu received the second project. The funders had no role in study design, data collection and analysis, decision to publish, or preparation of the manuscript.

**Competing interests:** The authors have declared that no competing interests exist.

applications [2]. It exhibits anti-inflammatory, antitumor, antifungal, antibacterial, and anxiolytic properties, presenting significant economic opportunities for farmers [3–5]. The substantial economic benefits associated with ginger cultivation have incentivized the continuation of monoculture. Consequently, this has led to a progressive increase in the prevalence of pathogens such as *Fusarium* wilt [6,7]. Owing to the scarcity of viable seeds and limited genetic diversity, breeding for disease resistance presents significant challenges [8]. Although certain resistant cultivars have been developed and implemented, *Fusarium* yellow continues to be prevalent across Asia, North America, and Oceania [9–11].

*Fusarium* wilt, caused by *Fusarium oxysporum* f. sp. *zingiberi* (*Foz*), was initially documented in Australia in the 1930s and is now prevalent in Asia, North America, and Oceania [7,9]. The pathogen is categorized within the phylum Ascomycota and the genus *Fusarium*, and it causes vascular diseases in a variety of plant species, including melon, cucumber, and potato [12–14]. The infection is initiated when the fungus infiltrates the host's vascular tissues through hyphal invasion, subsequently secreting enzymes that degrade the cell wall and producing mycotoxins [6,15]. Infected plants will exhibit chlorosis of their leaves, ultimately leading to the wilting and death of the entire above-ground portion of the plant. The wilted tissues will fall to the soil, where they will be decomposed by microorganisms, subsequently serving as potential sources for the accumulation of pathogens [16]. Research indicates that *Fusarium* spp. can persist saprophytically, enabling their survival in soil for prolonged durations in the absence of a host [17]. However, wilt manifests when the extent of root infection exceeds a critical threshold, thereby complicating control measures [18]. Concurrently, the activity and virulence of *Fusarium* spp. are also regulated by plant allelopathic compounds [19,20].

Allelopathy is a natural mechanism in plants that involves the release of chemicals, which affect the growth of other organisms [21]. Autotoxicity, a distinct form of intraspecific allelopathy, is characterized by the release of chemical compounds by plants that inhibit or suppress the germination and growth of conspecific individuals within the same species [22]. The allelochemicals released by the roots and decaying residues were isolated and identified, and were divided into different species based on their structures, including phenolic acids, aldehydes, coumarins, quinones, alkaloids, and terpenoids. Extensive research has been conducted on phenolic acids, which are believed to significantly contribute to the challenges encountered in continuous cropping systems [23]. These allelochemicals exert deleterious effects on seed germination, seedling development, root growth, plant quality, and resistance [24,25]. Increasing evidence also suggests that autotoxicity may initiate and modulate pathogen occurrence, thereby mediating interactions and signaling processes between plants and microorganisms [26,27]. For instance, cinnamic acids exuded by watermelon roots have been documented to inhibit seedling growth while concurrently augmenting the pathogenicity factors of *Fusarium oxysporum* f. sp. *niveum* [28]. Coumarins stimulated conidial germination and mycotoxin production of *Fon*, subsequently inducing a slowdown in plant growth [29,30]. The application of exogenous vanillic acid and p-hydroxybenzoic acid facilitates the spore germination of *F.*

solani and *F. equiseti*, while also modifying the structure of the rhizospheric soil microbial community [31]. These findings corroborate that plant root exudates facilitate wilt development.

Similar to other crops, ginger produces autotoxic leachates [32]. Although the phenomenon of autotoxic leachates is acknowledged, their precise role in the reproduction and virulence of pathogens remains unclear. Therefore, the four main components of ginger leachate from different organs, syringic acid, ferulic acid, coumarin and 7-Hydroxycoumarin, which have been verified to have negative effects on plant growth in our previous study [32], were employed, and their mechanisms of action were characterized. Changes in growth, reproduction and virulence factors of *Foz* were characterized.

## 2. Materials and methods

### 2.1. Pathogen strain and autotoxic compounds

The pathogenic strain JJF46 was isolated from infected ginger at the Laboratory of Plant Protection at the Weifang Institute of Technology, China. Morphological and molecular techniques were used to identify the pathogenic strain of *Fusarium oxysporum* f. sp. *zingiberi (Foz)*. According to a previous study and our preliminary results, four prevalent metabolites, syringic acid, ferulic acid, coumarin and 7-Hydroxycoumarin, which have been identified from the aqueous extract of ginger's organs and have negative effects on growth, were designated as autotoxic chemicals in the present study [32]. All chemicals were purchased from Aladdin Chemical Company (Shanghai, China).

Two distinct concentration gradients of autotoxic compounds were established based on the results of preliminary experiments. The first was used to assess the allelopathic effects of these compounds on mycelial growth (Experiment 1), whereas the second was designed to evaluate their effects on sporulation, conidial germination, and pathogenicity-related factors based on the differential sensitivity of *Foz* to the autotoxic compounds (Experiment 2). The final concentrations of autotoxins in Experiment 1 were 0, 0.05, 0.1, 0.25, and 0.5 mmol/L, while in Experiment 2, they were 0, 0.25, 0.5, 0.75, 1, and 2 mmol/L.

### 2.2. Measurement of *Foz* growth

To examine the impact of autotoxic compounds on mycelial growth, potato dextrose agar (PDA) was chosen as the medium for the administration of the autotoxic compounds. A mycelial plug with a diameter of 6 mm, derived from a 7-day-old pure culture, was positioned at the center of each plate and incubated at 28 °C for 6 days. Mycelial growth was assessed using the cross method after 2, 4, and 6d of incubation. The relative allelopathy intensity (RI) of mycelial growth was determined using Williamson's method [33]. Each treatment was performed in five replicates (n = 5).

### 2.3. Assessment of sporulation

To assess the allelopathic impact of autotoxic compounds on sporulation, five agar plugs, each 6 mm in diameter, were extracted from a 7-day-old PDA medium and inoculated into Bilay and Joffe's medium [34]. These were incubated at 28 °C for 7 days in 250 ml flasks. After the 7-day incubation period, the broth was filtered through four layers of sterile gauze, and 1 mL of the filtrate was serially diluted to concentrations ranging from $10^{-5}$ to $10^{-7}$ ind/ml. The resulting dilution was transferred to a hemocytometer for conidia enumeration using a microscope. Each treatment was conducted in quintuplicate (n = 5).

### 2.4. Measurement of conidial germination and biomass production

To investigate the impact of autotoxic compounds on conidial germination, *Foz* was cultured on PDA plates for seven days. Subsequently, five agar plugs, each 6 mm in diameter, were extracted from the culture and transferred to a liquid medium. The medium was incubated at 28 °C for seven days under shaking conditions at 180 rpm. The resulting broth was filtered through sterile gauze and diluted with sterile water to achieve a concentration of ≤ 1000 conidia/ml. A 100 µL

aliquot of the diluted suspension was spread onto PDA plates and incubated at 28 °C for two days. The number of colonies was counted to evaluate germination. Each treatment was performed in five replicates (n = 5).

The filtered fungal mycelia were collected on sterile filter paper and desiccated at 80 °C for 12 h until a constant mass was achieved. The corresponding filtrates were used for enzymatic assays. Each treatment was performed in five replicates (n = 5).

## 2.5. Determination of enzymes activity

To examine the influence of autotoxic compounds on enzymatic activity, five agar plugs obtained from a 7 day cultivation were transferred to a liquid medium and incubated at 28 °C for seven days under shaking conditions at 180 rpm. The resulting broth was filtered through four layers of sterile gauze and utilized as a crude enzyme preparation. Each treatment was performed in five replicates (n = 5).

Pectinase, cellulase, protease, and amylase were selected as cell wall-degrading enzymes (CWDEs) of the pathogen in this study. Pectinase activity was assayed as described by Chang et al. [35], with slight modifications. The reducing sugar released by enzyme hydrolysis of the polysaccharide substrate was reacted with 3,5-dinitrosalicylic acid (DNS) to produce a colored complex, which was quantified using an ultraviolet spectrophotometer. One unit of pectinase activity was defined as the amount of galacturonic acid (1 mg) produced by pectin breakdown per milliliter per hour. Cellulase activity was determined using the DNS method described by Brand and Alsanius [36], where one unit of activity was defined as the amount of enzyme catalyzing the release of 1 μg glucose per minute. Protease activity was measured following the method of Kole et al. [37], with one unit defined as the amount of enzyme catalyzing the hydrolysis that produced 1 nmol of tyrosine per milliliter of sample per minute. Total amylase activity was assessed using the method described by Murado et al. [38], where one unit of amylase activity was defined as the amount of enzyme that catalyzes the release of 1 mg of reducing sugar per mL of sample per minute.

## 2.6. Extraction and assay of mycotoxin

The response of mycotoxins to autotoxic compounds was assessed following the methodology outlined by Wu et al. [28], with slight modifications. The experiment commenced with the establishment of a standard curve using fusaric acid (Sigma Chemical Co.) at specified concentrations. Five agar plugs, each 6 mm in diameter, were extracted from 7-day-old PDA plates and inoculated into Richard's medium [39], followed by incubation at 28 °C for 30 days. Post-cultivation, the broth was filtered through sterile gauze, and the pH was adjusted to 2 using 2 mol/L HCl. The medium underwent five extractions with equal volumes of ethyl acetate, allowing the contents to settle for 30 min after each extraction. The organic phase was centrifuged and decolorized using activated charcoal. Subsequently, the filtrate was dried and condensed at 28 °C until a residue was obtained. The dried residue was reconstituted in 5 ml of ethyl acetate, and its optical density at 268 nm ($OD_{268}$) was measured using ultraviolet spectrophotometry. Each treatment was performed in five replicates (n = 5).

## 2.7. Measurement of expression of mycotoxin biosynthetic genes

To elucidate the allelopathic effects of autotoxic compounds on mycotoxin biosynthesis, the *FUB* gene cluster was selected for detailed examination and characterization [40]. This study included five genes from this cluster: *FUB1*, *FUB3*, *FUB6*, *FUB8*, and *FUB9*. The primer sequences and functional descriptions of these genes are listed in Table 1. Preliminary experiments indicated that reliable results of gene expression were obtained 7 days post-treatment. Therefore, a 6-mm-diameter agar plug obtained from a 7-day-old PDA culture was inoculated onto a fresh PDA plate supplemented with the corresponding autotoxic chemicals and incubated at 28 °C for 7 days. Subsequently, mycelia were harvested for gene expression analysis. Each treatment was performed in five replicates (n = 5).

**Table 1. Primer pair sequences and corresponding gene functions in the pathogen used for real-time PCR analysis.**

| Primer | Sequence (5'-3') | Predicted function |
|---|---|---|
| EF1α | F: 5'- ACCTCAATGAGTGCGTCGTCAC-3' | Translation elongation factor 1 alpha |
|  | R: 5'- CCCAGGCGTACTTGAAGGAACC-3' |  |
| FUB1 | F: 5'-TTCTTTGAGGCGCATGGAAC-3' | Polyketide synthase |
|  | R: 5'-TGTCTTGCCGAAAACGTTGC-3' |  |
| FUB3 | F: 5'-TGCTGCAGTCTTTGGAAAGC-3' | Aspartate kinase |
|  | R: 5'-ACCAAAACAGCGCACAGATC-3' |  |
| FUB6 | F: 5'-GCATCAAGCTCTGTTGGTGA-3' | NADP-dependent dehydrogenase |
|  | R: 5'-TCTGCACCAAGCTCATTGAC-3' |  |
| FUB8 | F: 5'-AAGGTTTGGTGCTTGAACCG-3' | Non-ribosome polypeptide synthase |
|  | R: 5'-ACGAACAAACTTGGCCTTGC-3' |  |
| FUB9 | F: 5'-AGCTGTCGTGTTGACTGTGG-3' | Flavin mononucleotide dependent dehydrogenase |
|  | R: 5'-GTCAGGAATACCGCCACCTA-3' |  |

Total RNA was extracted using a commercial MiniBEST Plant Total RNA Extraction Kit (TaKaRa, Dalian, China), verified by electrophoresis, and quantified using a NanoDrop spectrophotometer (Thermo Fisher, Germany). First-strand cDNA synthesis was performed using the Prime Script™ II 1st stand cDNA Synthesis kit (TaKaRa, Dalian, China). Quantitative real-time PCR (qRT-PCR) was performed using the TB Green™ Fast qPCR Mix kit (TaKaRa, Dalian, China) according to the manufacturer's instructions, as described by Li et al. [41]. Translation elongation factor 1 alpha (EF1α) was used as the internal reference [40]. Relative gene expression levels were calculated using the $2^{-\triangle\triangle Ct}$ method [42].

## 2.8. Statistical analysis

The allelopathic effects of autotoxic substances on fungal development, including mycelial growth, sporulation, colonial germination, and biomass production, were quantified using the relative intensity index (RI). The RI index was determined using the following equations: $RI = 1 - C/T$ ($T \geq C$) and $RI = T/C - 1$ ($T < C$), where C and T denote the control and treatment data, respectively. According to this index, a positive RI value ($RI > 0$) signifies stimulation, whereas a negative RI value ($RI < 0$) indicates inhibition [33]. To assess the allelopathic effects of autotoxic substances on cell wall-degrading enzymes (CWDEs), a weighted approach was employed for four enzymes based on their relative significance in the infection process [43]. The weights assigned to cellulase and pectinase were 0.3 each, while those assigned to protease and amylase were 0.2 each. The allelopathic effects of autotoxic chemicals on CWDEs were quantified using the equation: $PI = T/C \times W$, where C and T represent the control and treatment data, respectively. W denotes the weight, and PI represents the allelopathic intensity index. To comprehensively evaluate the allelopathic effects of autotoxic chemicals, the cumulative RI and PI indices were calculated as integrated measures.

All data were subjected to analysis using one-way ANOVA to determine the significance between the control and treatment groups. However, the amylase activity of 7-Hydroxycoumarin was assessed using multiple comparison tests. The significance level (P) was set at $P < 0.05$ and $P < 0.01$.

## 3. Results

### 3.1. Effect of autotoxins on mycelial growth and biomass production of *Foz*

The hyphal growth response exhibited variability among the autotoxic chemical. Mycelial growth remained stable, with a mean of $2.63 \pm 0.34$ (n = 65), under low-dose treatments (≤ 0.25 mmol/L) for all chemicals over a 2-day incubation period, except for syringic acid, which significantly enhanced fungal growth at concentrations ≤ 0.1 mmol/L (Fig 1A).

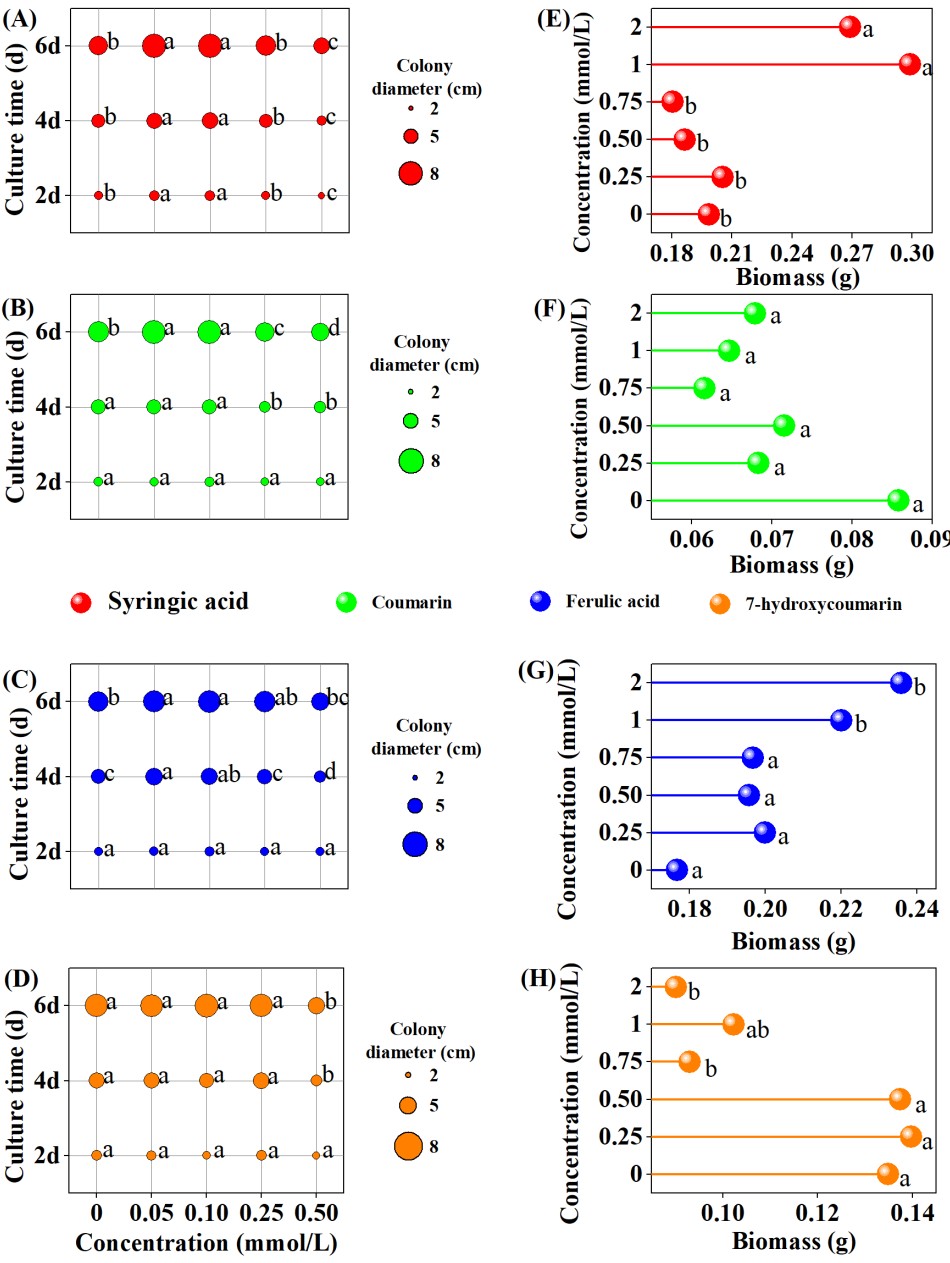

**Fig 1. Impact of various autotoxins on colony growth (A-D) and biomass production (E-H) of *Fusarium oxysporum* f. sp. *niveum* (*Foz*).** The autotoxins tested were **(A, E)**: Syringic acid; **(B, F)**: Coumarin, **(C, G)**: Ferulic acid, and **(D, H)**: 7-Hydroxyccoumarin. Different lowercase letters above bars indicate significant differences($P < 0.05$).

However, a marked inhibitory effect was observed at a higher concentration (0.5 mmol/L) of syringic acid. After 4 days of incubation, colony growth generally increased relative to the controls under low-dose treatments but was suppressed at higher concentrations, particularly with syringic acid, where the colony diameter was reduced by 30.36% at 0.5 mmol/L compared to the controls (Fig 1A). Consistent with the previous trend, fungal growth was significantly stimulated by low-dose treatments of all autotoxic chemicals except 7-Hydroxycoumarin over a 6-day incubation period (Fig 1D). The

most pronounced stimulatory effect was observed with syringic acid at 0.05 mmol/L, where the colony diameter reached 7.83 cm after 6 d of incubation. In contrast, the high-dose treatment inhibited mycelial growth, corroborating the results observed after 4 days of exposure. At a coumarin concentration of 0.5 mmol/L, the colony diameter was reduced to 4.82 cm compared to 6.38 cm in the control (Fig 1B).

Biomass production did not exhibit the same pattern as mycelial growth. Coumarin had no significant impact on biomass accumulation, with the dry weight of the mycelia remaining constant at 0.07 ± 0.01 g (Fig 1F). At lower concentrations of the other chemicals, no significant changes were observed after 7 days of incubation; however, distinct responses were observed at higher concentrations. Ferulic and syringic acids significantly increased fungal biomass compared to the controls by 33.45% and 35.59%, respectively, at concentrations ≥1 mmol/L (Fig 1E and G). Conversely, 7-Hydroxycoumarin demonstrated an inhibitory effect, as evidenced by a marked reduction in biomass at concentrations ≥0.75 mmol/L (Fig 1H).

### 3.2. Effect of autotoxins on sporulation and spore germination of *Foz*

Sporulation responses exhibited significant variation across the different chemical treatments. At low concentrations (≤ 0.75 mmol/L), coumarin had minimal influence on sporulation (Fig 2B). However, with increasing coumarin concentrations, spore production was markedly enhanced, increasing from $38.67 \times 10^6$ in the control to $90 \times 10^6$ at 2 mmol/L. A similar positive effect was observed with ferulic acid (Fig 2C), where sporulation increased from $44.33 \times 10^6$ in the control to $72.67 \times 10^6$ at 1 mmol/L, although this stimulation diminished at higher concentrations. The effects of syringic acid (Fig 2A) and 7-Hydroxycoumarin followed a comparable pattern: sporulation was significantly promoted at low concentrations ≤ 0.75 mmol/L, but declined at higher concentrations (2 mmol/L). At the highest concentration, spore numbers in the liquid medium decreased to $7.37 \times 10^6$ and $3.73 \times 10^6$ for syringic acid and 7-Hydroxycoumarin, respectively (Fig 2A and D). Ferulic acid (Fig 2G) did not induce significant changes in spore germination, with colony numbers remaining relatively stable at $83.97 \pm 2.67 \times 10^6$ (n = 30). A slight increase was observed in the coumarin treatment at 0.75 and 1 mmol/L (Fig 2F). In contrast, syringic acid (Fig 2E) and 7-Hydroxycoumarin (Fig 2H) significantly enhanced spore germination at low concentration. However, both compounds exhibited mild inhibitory effects at their highest concentrations. Specifically, spore germination under syringic acid declined from $74.50 \times 10^6$ in the control to $62.25 \times 10^6$ at 2 mmol/L, while under 7-Hydroxycoumarin, it decreased from $99.00 \times 10^6$ in the control to $73.50 \times 10^6$ at 2 mmol/L.

### 3.3. Impact of autotoxins on the activity of cell-wall degrading enzymes of *Foz*

After 7 days of exposure to varying concentrations of autotoxic compounds, pectinase activity showed distinct responses to each treatment (Fig 3A). In the presence of ferulic acid, enzyme activity was significantly inhibited compared with the control by 65.89%, 52.64%, 49.52%, 75.61%, and 30.50% at concentrations of 0.25–2 mmol/L, showing strong, concentration-dependent inhibition. In contrast, both coumarin and syringic acid markedly stimulated pectinase activity at concentrations ≥0.75 mmol/L, with syringic acid having the most pronounced effect. For 7-Hydroxycoumarin, pectinase activity initially increased from 0.45 mg/mL/h at 0.25 mmol/L to 0.57 mg/mL/h at 0.75 mmol/L, but subsequently declined with higher doses, reaching a minimum of 0.21 mg/mL/h at the highest concentration.

In the presence of syringic acid (Fig 3B), amylase activity was significantly stimulated in a concentration-dependent manner, increasing from 0.12 to 0.21 mg/mL/min across all treatment levels. Similar positive effects were observed under 7-Hydroxycoumarin and coumarin treatments (Fig 3B) at concentrations ranging from 0.5 to 2 mmol/L, although a slight decline was detected at the highest concentration of 7-Hydroxycoumarin. In contrast, ferulic acid (Fig 3B) exhibited a dose-dependent effect, significantly enhancing amylase activity at the lowest concentration and inhibiting it at higher concentrations.

Syringic acid significantly induced cellulase activity (Fig 3C), a key enzyme in pathogen infection, although not in a concentration-dependent manner. Activity remained consistently elevated, averaging 61.35 µg/ml/min (n = 15). Ferulic acid

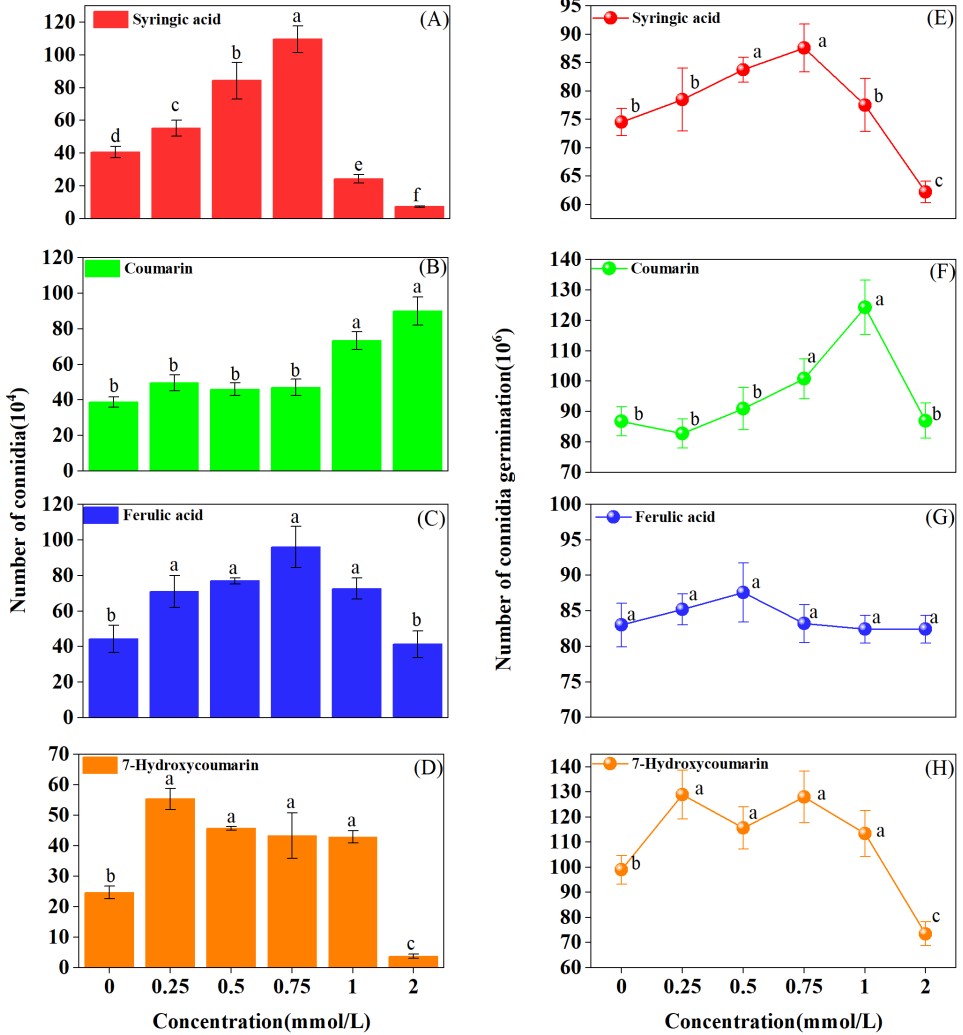

**Fig 2. Effects of different autotoxins concentrations on sporulation and conidial germination of *Fusarium oxysporum f. sp. niveum (Foz)*.** Effects on sporulation by **(A)**: Syringic acid **(B)**: Coumarin **(C)**: Ferulic acid and **(D)**: 7-Hydroxyccoumarin. Effects on conidial germination by **(E)**: Syringic acid **(F)**: Coumarin **(G)**: Ferulic acid **(H)**: 7-Hydroxyccoumarin. Different lowercase letters above bars indicate significant differences (*P* < 0.05).

and 7-Hydroxycoumarin had minimal effects (Fig 3C), with only a slight increase in activity observed at concentrations ≥2 mmol/L and ≥1 mmol/ L, respectively.

Analogous to other enzymatic responses, protease activity in syringic acid (Fig 3D) treatments (≥0.75 mmol/L) increased significantly from 13.12 nmol/mL/min in the control treatment to 63.32 nmol/mL/min at 2 mmol/L. Similarly, protease activity was markedly enhanced in coumarin treatments (≥ 0.75 mmol/L), showing a 377.66% increase at 2 mmol/L (Fig 3D). In contrast, ferulic acid (Fig 3D) promoted protease activity at low concentrations (≤ 0.75 mmol/L) but significantly inhibited it at higher concentrations (37.86% at 2 mmol/L). No significant changes were detected in coumarin treatments, and enzyme activity remained within the range of 12.49 ± 3.04 nmol/mL/min (Fig 3D).

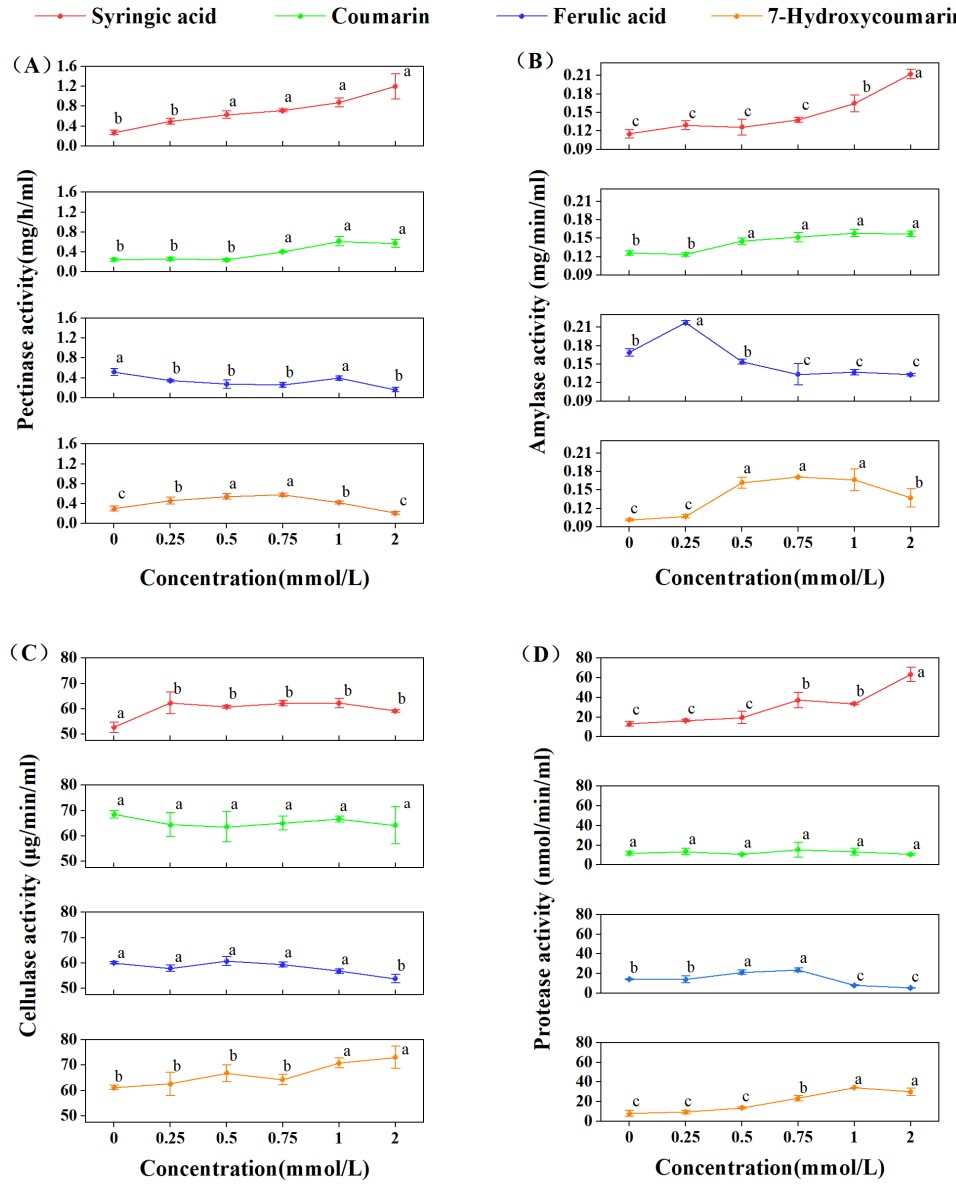

**Fig 3. Effects of autotoxins at different concentrations on activity of cell-wall degrading enzymes. (A)**: Pectinase activity; **(B)**: Amylase activity; **(C)**: Cellulase activity; **(D)**: Protease activity. Different lowercase letters indicate significant differences at ($P$<0.05).

### 3.4. Effect of autotoxins on mycotoxin production of *Foz*

Mycotoxin production was strongly stimulated by most autotoxins in liquid culture, with the exception of 7-Hydroxycoumarin which exerted an inhibitory effect (Fig 4). The stimulatory effects of syringic acid and coumarin were found to be concentration-dependent (Fig 4A). At their highest concentrations, syringic acid and coumarin yielded highest concentration syringic acid and coumarin resulted in mycotoxin levels of 36.98 µg/ml and 38.56 µg/ml, corresponding to 2.99 and 3.28 fold increases over the control, respectively. A similar promotive effect was observed for ferulic acid (Fig

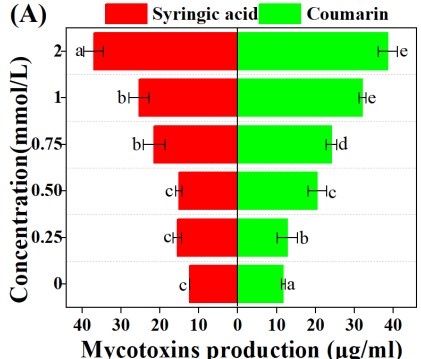
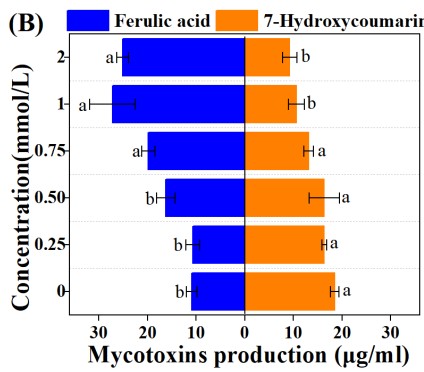

**Fig 4. Mycotoxin production in response to different concentrations of autotoxins. (A)**: Syringic acid and Coumarin **(B)**: Ferulic acid and 7-Hydroxycoumarin Different lowercase letters indicate significant differences among treatments ($P<0.05$).

[4B](), which increased mycotoxin production from 10.90 µg/ml to 27.19 µg/ml. Conversely, 7-Hydroxycoumarin progressively reduced mycotoxin yields with increasing concentration ([Fig 4B]()).

### 3.5. Impact of autotoxins on the expression of mycotoxin biosynthetic genes

The expression of mycotoxin biosynthetic genes varied under different autotoxin treatments. Syringic acid ([Fig 5A]()) markedly induced mycotoxin accumulation, as evidenced by the gradual upregulation of *FUB3* and *FUB9*, which increased by approximately twofold relative to the control. In contrast, the relative expression levels of *FUB1*, *FUB6* and *FUB8*, showed only slight increases in response to syringic acid. Coumarin ([Fig 5B]()) strongly upregulated *the FUB* gene cluster, particularly *FUB3*, *FUB6*, and *FUB9*, at the highest treatment concentration (2 mmol/L), showing 2.41-, 2.64-, and 2.90-fold increases compared to the control, respectively. *FUB1* and *FUB8* expression levels also increased at coumarin concentrations ≥1 mmol/L. In the presence of ferulic acid ([Fig 5C]()), the relative expression of *FUB9* initially increased significantly at lower concentrations (≤1 mmol/L) and then declined to 2.43-fold of the control at the highest concentration. At 0.75 mmol/L and 1 mmol/L, *FUB9* expression reached 5.43-and 6.31- fold of the control, respectively. A similar trend was observed *for FUB6* and *FUB8*. The *FUB3* gene displayed a comparable response, although the stimulatory effect was weakened at 1 mmol/L. In 7-Hydroxycourmarin treatments ([Fig 5D]()), there was a reduction in the expression of *FUB6* declined from 0.96 at 0.25 mmol/L to 0.77 at 2 mmol/L while *FUB9* expression decreased sharply, reaching its lowest level (0.52) at 0.75 mmol/L. *FUB3* and *FUB8* expression remained largely unchanged, and *FUB1* showed only slight induction in response to 7-Hydroxycourmarin.

### 3.6. Comprehensive effect of different autotoxins on *Foz*

The allelopathic effect of autotoxins on *Foz* varied distinctly ([Fig 6]()). Syringic acid was the most comprehensive stimulant, strongly enhancing all measured aspects of pathogenicity, particularly mycelial growth and CWDEs. The RI indices were 0.24 and 1.88, respectively. Coumarin strongly induced mycotoxin synthesis but concurrently reduced mycelial growth and biomass and had a slight inhibitory effect on CWDEs. Ferulic acid significantly increased sporulation and biomass but had a slight inhibitory effect on CWDEs. Conversely, 7-Hydroxycoumarin had the most divergent effects; it was the strongest enhancer of conidial germination but also the strongest inhibitor of mycelial growth, biomass accumulation, and mycotoxin production.

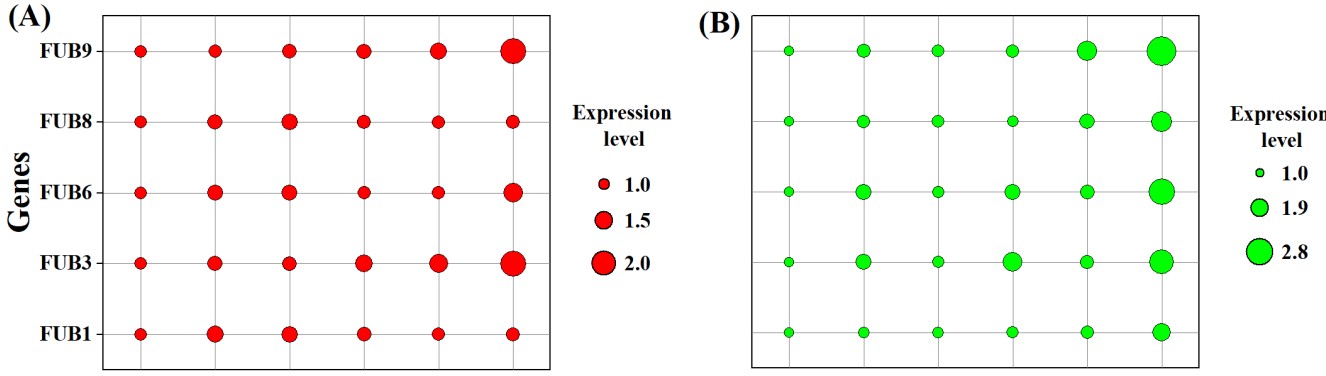

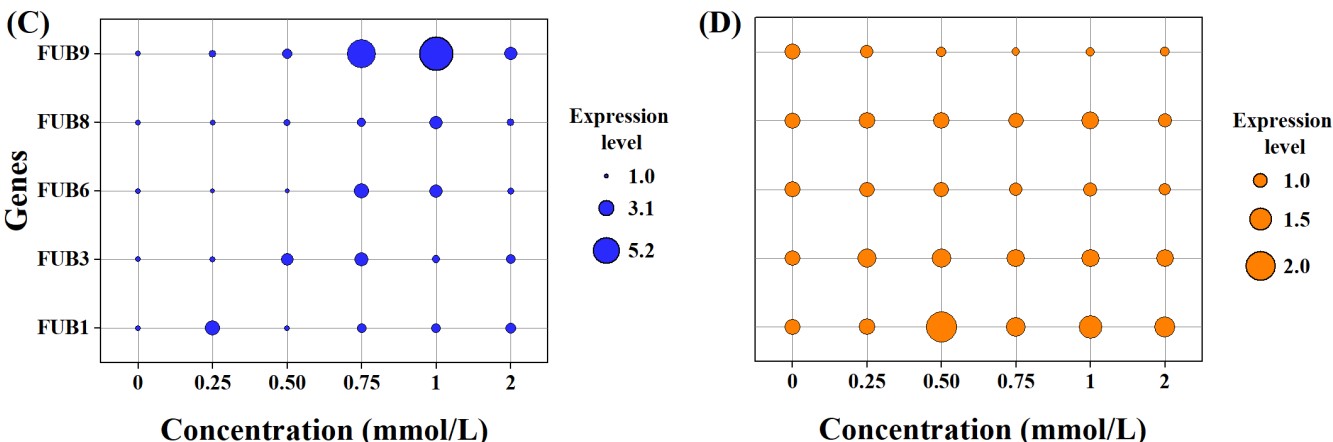

**Fig 5. Effect of different autotoxins concentrations on the expression of *FUB* gene cluster of *Fusarium oxysporum* f. sp. *niveum (Foz).* (A):** Syringic acid, **(B)**: Coumarin, **(C)**: Ferulic acid, (D) 7-Hydroxycoumarin.

## 4. Discussion

### 4.1. The allelopathic effect of autotoxins on *Foz* as a consequence of the interaction between ginger's metabolites and pathogens

Autotoxicity, a prevalent natural phenomenon, is recognized as a significant factor contributing to soil sickness in numerous crops [44]. Various autotoxic compounds have been identified that exert substantial inhibitory effects on plant growth and yield, primarily by exacerbating soil-borne diseases. Phenolic acids, coumarins, and esters have been identified as the principal autotoxins [45–47]. In the context of ginger, previous research has demonstrated that ginger root exudates markedly suppress seedling growth, impair antioxidant enzyme function, and alter membrane permeability [32]. The present study elucidated the allelopathic effects of autotoxins on pathogenic fungi. These findings indicate that the four chemicals exhibited distinct effects on *Fusarium* wilt in ginger, likely attributable to variations in their molecular structures. At specific concentrations, these compounds affected mycelial growth, spore germination, cell wall-degrading enzyme activity, and mycotoxin production (Fig 7).

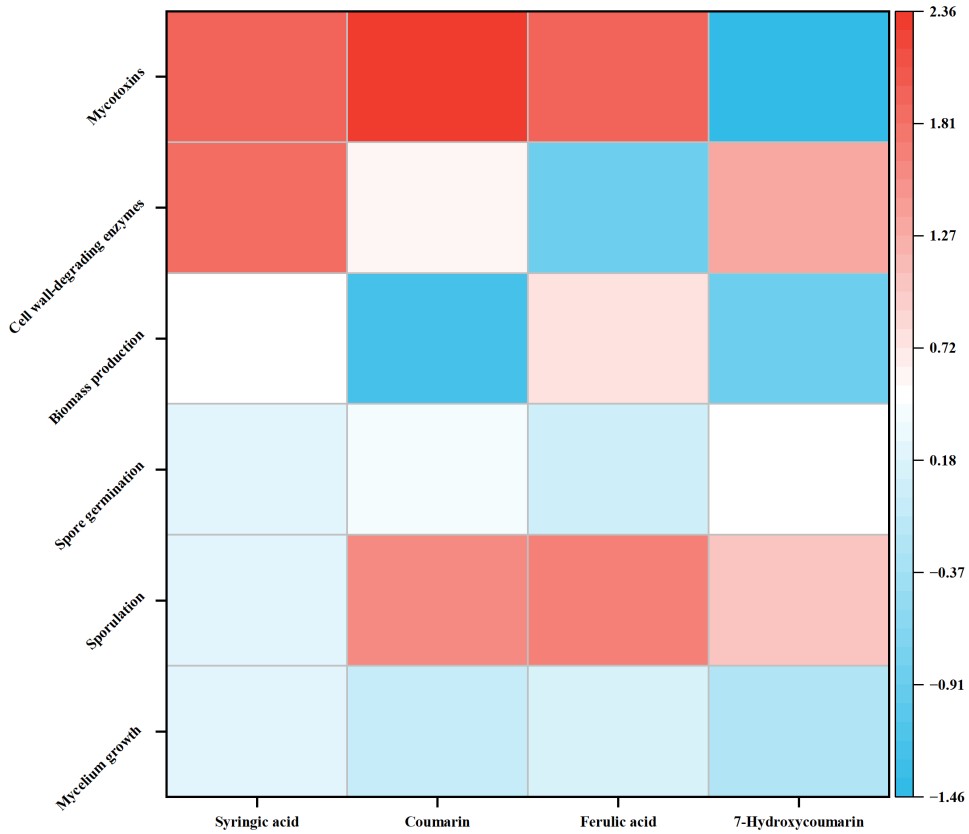

**Fig 6. Comprehensive effect of different autotoxin on *Fusarium oxysporum* f. sp. *niveum* (*Foz*).**

*Fusarium oxysporum* f. sp. *zingiberi* (*Foz*) is a common soil-borne pathogen that causes ginger wilt, a disease characterized by leaf withering and plant wilting [9]. The molecular mechanisms underlying *Foz* infection have been widely studied [48]. Under natural conditions, this soil-borne disease typically progresses through several phases. The initial phase is often triggered by root exudates from the host plant, such as phenolic and amino acids, which signal the pathogen to initiate its invasion program by stimulating spore germination and germ tube growth [49]. Second, the germ tube of the pathogen colonizes the root surface, establishing the first physical contact [50]. It then attaches to and forms a mycelial network that proliferates throughout the host root system. In the third phase, the fungus secretes CWDEs to break down the host's defenses and invade the root cortex and vascular tissues, including the xylem vessels [51]. Finally, pathogens release toxins and other virulence factors to induce disease symptoms in the host plant [15]. Therefore, to elucidate the role of autotoxic chemicals in promoting plant disease incidents, it is essential to first interpret the allelopathic effects of these chemicals on the physiological activities of pathogens. Previous research has established that plant metabolites, integral components of the immune system, have evolved a sophisticated and effective defense mechanism against various phytopathogenic infections [52–54]. Within this framework, numerous metabolites have been shown to inhibit the growth, reproduction, and pathogenicity of plant pathogens at specific concentrations, suggesting that the synthesis of these compounds constitutes a defensive strategy. Methyl jasmonate, secreted by *Citrullus lanatus*, can induce resistance in watermelon and inhibit the infection of *F. oxysporum* f. sp. *Niveum* race 2 (*Fon* 2) by upregulating the expression of the melatonin biosynthetic gene *caffeic acid O-methyltransferase 1* (ClCOMT1) and enhancing melatonin accumulation [55]. The hordedane diterpenoid produced by barley has been found to inhibit the colonization of *F. graminearum*, attributed to

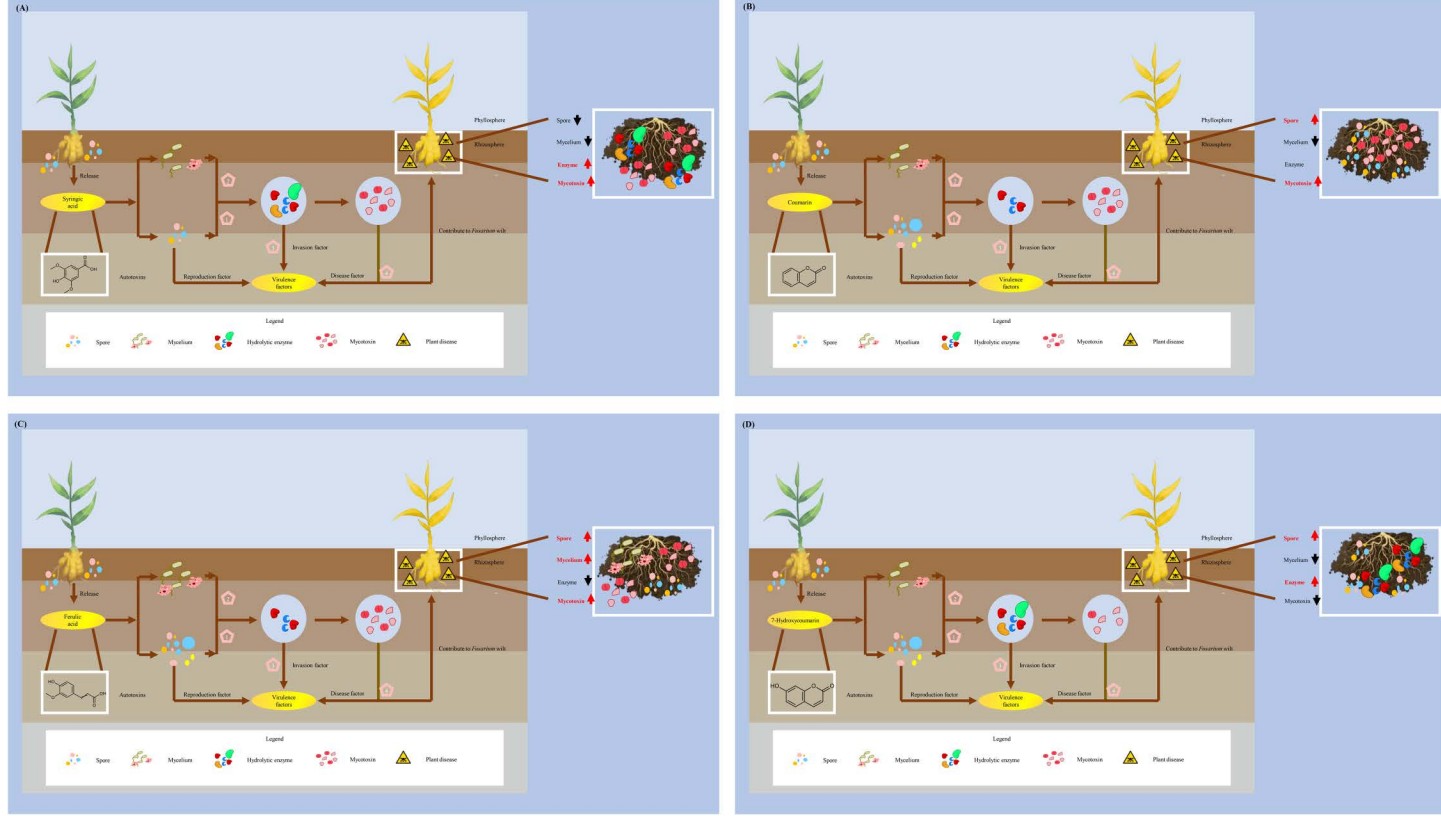

**Fig 7. Schematic diagram of effect of different autotoxins on *Fusarium oxysporum* f. sp. *niveum* (*Foz*). (A)**: Syringic acid **(B)**: Coumarin **(C)**: Ferulic acid; **(D)**: 7-Hyoxycoumarin.

the expression of diterpene synthase gene clusters [56]. Consequently, we posit that the impact of ginger metabolites on *Foz* is a result of the regulation of plant metabolism.

## 4.2. Ginger metabolites regulate the growth and reproduction of *Foz*

In a natural environment, the growth and reproduction of plant pathogens within the host represent their fitness and survival capacity [57]. These physiological processes are regulated not only by the pathogens themselves but also by root exudates secreted by the host plants. In the present study, most autotoxic chemicals followed a typical hormetic response, except 7-Hyoxycoumarin, exhibited similar effects on mycelial growth, where no significant variation was recorded at low concentrations but high concentrations (≥0.25 mmol/L) inhibited mycelial growth (Fig 1), consistent with previous reports [58]. Over several years of cultivation, plant metabolites have progressively accumulated in the soil, reaching a concentration of 0.25 mmol/L. Consequently, the allelopathic effect of autotoxins transitions from stimulation to inhibition, which is regarded as one of the plant's defense mechanisms against pathogens [55,59]. Meanwhile, the stimulatory effect of the other phenolic acids diminished over time, as indicated by the reduction in the mycelial growth rate on solid medium (Fig 1). This phenomenon may be attributed to the fact that *Foz* acclimated to the hormesis effect induced by ginger's metabolites. Previous research has shown that numerous plants regulate the growth and reproduction of soil-borne pathogens by altering the quantity and composition of organic acids in their root exudates [60–62]. Therefore, the diminished promotive effect of these chemicals may represent that *Foz* has adapted the regulation of ginger. However, the effect on biomass

was inconsistent with that on mycelial growth, aligning with findings from a previous study [63]. The dry weight of *Foz* comprises mycelia, spores, and their cellular constituents. Under typical environmental conditions, the growth of mycelia, spore production, and the synthesis of organic macromolecules collectively contribute to biomass accumulation. However, when phytopathogens are subjected to abiotic stress, they accumulate proteins and enzymes to bolster their resistance to such stress [64]. Consequently, the observed discrepancy between mycelial growth and biomass accumulation may be attributed to the synthesis of organic macromolecules. Furthermore, the differing conditions of plate and liquid cultures may also elicit distinct physiological responses in *Foz* to counteract the allelopathic effect [63]. In present study, ferulic and syringic acids increased biomass accumulation at high concentrations (> 1 mmol/L), coumarin and 7-Hydroxycoumarin significantly inhibited it (Fig 1). Specifically, phenolic acids (i.e., ferulic acid and syringic acid) may have acted as additional carbohydrate sources for the pathogen, thereby stimulating biomass production [65]. However, coumarins compromised the structural integrity of cells, increased mycelial conductivity, and induced extracellular protein leakage, ultimately leading to a reduction in *Foz* biomass [66].

Spores are essential reproductive units in fungi [67]. Previous research has established a correlation between spore content and the pathogenicity of *Fusarium* spp. [15]. In the present study, autotoxic chemicals exhibited a consistent pattern of sporulation and colonial germination, characterized by stimulation at low concentrations and inhibition at high concentrations, with the exception of coumarin and ferulic acid (Fig 2). The 7-Hydroxycoumarin exhibited a clear concentration-dependent response. This phenomenon may explain the increased incidence of *Fusarium* wilt in ginger cultivated in monoculture systems [26]. Concurrently, this promotion may be driven by the pathogen's capacity to remodel host metabolism and induce the production of metabolites that facilitate pathogen reproduction [68]. Moreover, high coumarin concentrations notably stimulated sporulation and conidial germination (Fig 2B and F), corroborating findings from previous research [29]. The observed response may be attributed to the regulation of *Foz* by coumarins. Previous research has documented that plant metabolites can modulate the gene expression of phytopathogens, thereby influencing their activity [69,70]. Consequently, genes associated with aerobic respiration and ATP synthesis, such as *FoSir* 5, which governs the initiation of aerobic respiration, may be impacted by coumarin. This interaction ultimately stimulates spore production and germination [69].

### 4.3. *Foz* modulates pathogenicity factors in response to the regulatory effects of ginger

Cell wall-degrading enzymes (CWDEs) are critical virulence factors in plant-pathogenic fungi [43]. In *Fusarium oxysporum*, the enzymatic activities of pectinases and cellulases are integral to the degradation of pectin and cellulose, thereby facilitating host invasion [71,72]. During colonization, pectinases are initially induced and expressed to facilitate pectin degradation by removing methoxy ester groups [73]. Following pectin de-esterification, other cell wall-degrading enzymes can act more effectively on cell wall components [74]. Previous research has also identified amylases and proteases as pathogenicity factors, with gene knockouts resulting in the total or partial loss of pathogenicity in *F. oxysporum* [75,76]. In the present study, syringic acid (Fig 3) exhibited the most pronounced stimulatory effect on these hydrolytic enzymes, even at low concentrations (≥0.25 mmol/L), which is consistent with previous studies [30,77]. Ferulic acid also enhanced pectinase and amylase activities at specific concentrations. During pathogen infection, plants may accumulate substantial amounts of phenolic acids, which are regarded as crucial components of plant resistance, inhibiting pathogen growth at specific concentrations [78,79]. Consequently, the stimulatory effect of the phenolic acid may represent the response of *Foz* to the regulatory mechanisms of ginger. A substantial concentration of syringic acid inhibited the mycelial growth and reproduction of *Foz* (Fig 1); however, the pathogens may upregulate the expression of pathogenicity-related genes, such as *pl1* and *pme*, which encode the CWDEs, to adapt to host regulation [80]. The increased activity of these enzymes aids *F. oxysporum* in hydrolyzing polymers and compromising the integrity of plant cell wall polymer. Additionally, coumarin and 7-Hydroxycoumarin also stimulated CWDEs activity at specific concentrations, corroborating previous findings [29]. Nevertheless, this phenomenon is intricate due to the

substantial disparity between the PDA medium and the natural soil environment. This apparent contradiction under-scores the context-dependent influence of phenolic compounds, which can either promote or suppress the secretion, synthesis, and activity of CWDEs.

The secretion of secondary metabolites by fungal plant pathogens is essential for their survival and adaptation to dynamic environments. *Fusarium* spp., which are model plant pathogens, can produce a range of mycotoxins that act as critical virulence factors or effectors, leading to plant wilting [81]. Among these, fusaric acid (FA), also known as 5-butylpicolinic acid, is a polyketide-derived secondary metabolite commonly synthesized by *Fusarium* species and is closely associated with wilt symptoms in host plants [48]. The phytotoxicity of FA is dependent on its concentration; moderate doses significantly inhibit plant growth and reduce biomass, whereas high concentrations result in severe leaf wilting and necrosis [47]. The biosynthesis of fusaric acid is regulated by a gene cluster con-sisting of 12 genes, collectively referred to as the *FUB* cluster, whose expression directly affects FA production [62]. Previous studies have identified *FUB1*, *FUB3*, and *FUB6* as essential genes involved in FA synthesis [40]. In the present study, the majority of autotoxic compounds demonstrated a pronounced stimulatory effect on FA in *Fusarium oxysporum* f. sp. *zingiberi* (*Foz*), with the exception of 7-Hydroxycoumarin (Fig 4). However, the mechanisms underlying stimulation by autotoxic compounds appear to vary, as evidenced by distinct gene expression patterns (Fig 5). Syringic acid markedly induced mycotoxin accumulation, as evidenced by the gradual upregulation of *FUB3* and *FUB9*, which increased by approximately twofold relative to the control. Coumarin exhibited the most substan-tial stimulatory effect on FA synthesis, accompanied by the upregulation of *FUB3*, *FUB6*, and *FUB9*. These findings suggest that coumarin may enhance the activity of amino acid kinase and NADH-dependent dehydrogenase in the FA biosynthetic pathway [40]. However, FA production was significantly diminished under 7-Hydroxycoumarin treatments, likely due to the inhibition of *FUB6* expression, suggesting that 7-Hydroxycoumarin may mitigate the phytotoxicity of *Foz*. In natural ecosystems, coumarin and 7-hydroxycoumarin, which are representative pheno-lic metabolites of plants, are typically characterized as defense compounds due to their inhibitory effects on the growth and pathogenicity of pathogens [82–84]. Therefore, the stimulatory effect of coumarin should be attributed to the responses of *Foz* to ginger, which is analogous to the response of CWDEs. In treatments with ferulic acid, significant upregulation of *FUB1* and *FUB9* was observed, indicating that ferulic acid may augment the activity of polyketide synthase.

### 4.4. A comprehensive understanding of the interactions between ginger metabolites and *Foz*

Plant metabolites are recognized as primary defense mechanisms against soil-borne pathogens, influencing micro-bial composition and function. Numerous studies have demonstrated that plant metabolites can impact various physiological activities of plant pathogens, leading to diverse effects on these pathogens [60,85]. In present study, RI indices were employed to offer a comprehensive understanding of this interaction. According to the RI indices, syringic acid significantly stimulated mycelial growth and the activity of cell wall-degrading enzymes, while exerting a slight negative effect on growth and reproduction (Fig 6). Coumarin strongly induced mycotoxin synthesis and sporu-lation but inhibited biomass accumulation (Fig 6). Ferulic acid notably enhanced sporulation and mycotoxin synthesis but markedly decreased CWDE activity (Fig 6). A similar phenomenon was observed with 7-Hydroxycoumarin, which facilitated sporulation and the infection program but inhibited mycelial growth, biomass production, and mycotoxin synthesis (Fig 6). The variation in the allelopathic effects of autotoxins on *Foz* is attributed to the distinct action sites of different compounds, and phytopathogens subsequently respond to the regulation of plants by strengthening their pathogenicity to break down the plant defense systems. (Fig 7). However, it is important to note the significant dif-ferences between the PDA medium and the natural soil environment. Consequently, the present study may not fully reflect the actual interaction patterns between plant metabolites and phytopathogens, and future research should focus on their interactions in field conditions.

## 5. Conclusions

In this study, four autotoxic compounds, syringic acid, ferulic acid, coumarin, and 7-hydrocoumarin, were selected as representative agents to investigate the interactions between ginger metabolites and *Fusarium oxysporum* f. sp. *zingiberi* (*Foz*) at varying concentrations. Syringic acid significantly inhibited mycelial growth, sporulation, and spore germination, while markedly enhancing the activity of cell wall-degrading enzymes and mycotoxin synthesis. Coumarin exhibited a pronounced inhibitory effect on biomass production, concurrently stimulating sporulation and mycotoxin synthesis. Ferulic acid treatment reduced the activity of cell wall-degrading enzymes and spore germination, while upregulating sporulation and mycotoxin synthesis. A similar pattern was observed with 7-hydrocoumarin, which exerted a strong inhibitory effect on mycelial growth, biomass production, and mycotoxin synthesis, although it slightly stimulated the activity of cell wall-degrading enzymes and sporulation. Our findings provide the first evidence of the interaction between ginger autotoxins and *Foz*. This study lays a theoretical foundation for further in-depth investigations into the mechanisms underlying the effects of autotoxic substances on *Foz* and the mitigation of continuous cropping-related disorders.

## Supporting information

**S1 Table. The preliminary data for colony growth and biomass production.**
(XLSX)

**S2 Table. The preliminary data for spore germination and sporulation.**
(XLSX)

**S3 Table. The preliminary data for cell-wall degrading enzymes.**
(XLSX)

**S4 Table. The preliminary data for mycotoxins production.**
(XLSX)

**S5 Table. The preliminary data for gene expression.**
(XLSX)

**S6 Table. The preliminary data for comprehensive effect of different autotoxin on *Foz*.**
(XLSX)

## Author contributions

**Conceptualization:** Yanping Xu, Naicheng Li, Abdullah Gera.

**Formal analysis:** Yan Zhang.

**Funding acquisition:** Naicheng Li.

**Investigation:** Yan Zhang.

**Methodology:** Yan Zhang, Hongshen Guo, Xiaochuan Chen, Miaomiao Zhang.

**Resources:** Hongshen Guo.

**Software:** Xiaochuan Chen.

**Validation:** Miaomiao Zhang, Naicheng Li.

**Visualization:** Naicheng Li.

**Writing – original draft:** Yan Zhang.

**Writing – review & editing:** Abdullah Gera.

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
