## [Decision Letter · Decision Letter 0]

30 Mar 2026

PONE-D-25-63719In vitro study of the growth, reproduction and pathogenicity responses of Fusarium oxysporum f. sp. zingiberi to autotoxins from gingerPLOS One

Dear Dr. Naicheng,

Thank you for submitting your manuscript to PLOS ONE. After careful consideration, we feel that it has merit but does not fully meet PLOS ONE’s publication criteria as it currently stands. Therefore, we invite you to submit a revised version of the manuscript that addresses the points raised during the review process.

We look forward to receiving your revised manuscript.

Kind regards,

Eugenio Llorens

Academic Editor

PLOS One

Journal Requirements:

“This work was supported by grants from the Shandong Provincial Natural Science Foundation (grant number: ZR2023QC326) and Key R&D Program of Shandong Province，China (grant number: 2023TZXD028). Li received the first project and Xu received the second project.”

4. We notice that your supplementary figures are uploaded with the file type 'Figure'. Please amend the file type to 'Supporting Information'. Please ensure that each Supporting Information file has a legend listed in the manuscript after the references list.

Reviewer's Responses to Questions

**Comments to the Author**

1. Is the manuscript technically sound, and do the data support the conclusions?

Reviewer #1: No

Reviewer #2: Partly

2. Has the statistical analysis been performed appropriately and rigorously? 

Reviewer #1: Yes

Reviewer #2: Yes

3. Have the authors made all data underlying the findings in their manuscript fully available?

Reviewer #1: Yes

Reviewer #2: Yes

4. Is the manuscript presented in an intelligible fashion and written in standard English?

Reviewer #1: Yes

Reviewer #2: Yes

5. Review Comments to the Author

Reviewer #1: Dear authors,

The article entitled “In vitro study of the growth, reproduction and pathogenicity responses of Fusarium oxysporum f. sp. zingiberi to autotoxins from ginger” resulted to me quite interesting well written but, the conclusions does not fit with the rsults presented and I am not according to them, this makes me to suggesting several poinst that might be improved after an extensive review of content, conclusions, discussion and some changes that might be useful to improve the scientific impact and general quality of the article and conclusions obtained.

Below I detailed my suggestions waiting that it will be useful.

Sincerely

Title

I would focus the title adjusting it to the main conclusions of the article, from my point of view I see much more relevant the founding about the effects of the different metabolites into the fungal metabolism so I would suggest this kind of title or similar:

Physiologic and molecular response of Fusarium oxysporum f. sp. zingiberi to ginger specific metabolites.”

I would suggest to author to review first my suggestion to understand this proposed title, based principally into the quite differences found between the effects produced by the different plant metabolites, more than the term autotoxins, considering that the specific effects of plant metabolites against high impact phytopathogens is poor and scarce known compared to other areas of knowledge.

Abstract

Considering making a big change into the structure and content of the abstract, check line 68 to 75 to present the results under this point of view, add all detailed results including FUB3 and FUB9 not included, highlight the role of cumarin but consider that it might be possible that the fungus is defending from cumarin effect, trying to not death and this might be the reason because you see the inhibition of mycelial growth but strong mycotoxin synthesis, because in plant the fungus try to eat death tissues and it likes a defense response of the fungus against coumarin more than that this metabolite acts as an autotoxin to promote Foz.

Explain also at the abstract in which the autotoxins are representative, but I would make a great effort for differentiate what is a fungal defense response against toxic and plant-defense-metabolites such as coumarin and derivates and which produce an effect of real autotoxin, which does not fit with current plant defense responses.

Lines 28-31.- make a fussion and shorter sentence which is repetitive.

Line 34-39.- consider change the conclusions accordingly to results.

Introduction

Lines 53-54.- detail, is there varieties, ecotypes with different resistances? Explain here.

Line 62.- cite roots here.

Lines 68-75.- consider add this part to abstract.

Line 82.- explain why coumarin inhibit plant growth? Or is just that coumarin is produced in response to the pathogen and plant stop the growth until defense mechanism are activated? Justify. Currently coumarin is produced as plant-defense metabolite.

Line 92.- the term “autotoxic exudades” means that the plant produces spontaneous cell death and currently this is associated with hypersensitive response, so I would try to present this as and HR produced in response for defense not to be “autotoxic”, the death with pathogens only has sense when this death stop the pathogen and save the plant.

Lines 95-98.- cite here why you used and selected them, why they come from leaves (see methods) and explain previous specific effects of all of them, here it is necessary to add the steps in wich those metabolites are produced by the plant during defense responses to Foz, check previous works including other plant systems and metabolomic studies made of plant systems responding to Foz and against Fol also and determine the sequence and kinetic about how those metabolites are produced by the plant. Probably the kinetic s coordinated and correlated with fungal vital cycle of infection and with penetration into the root.

Those points has to be considered along the work, as well as into results and conclusion and discussion section. Each metabolite has and specific effect on the fungus because attach the fungus at the different stages of plant-fungal colonization.

Line 106.- at this article indicates studies in leaves explain and justify in methods why use them.

Line 112.- Transform this table into text shorter because in all of them the times are the same.

Line 118.- add as follows:

Line 122.- why PDA ? how PDA interacts with metabolite structure? Did you check at controls?

Lines 156-173.- Did you check if inductions of those plant-genes CWDEs are related to production of certain metabolites ??? check and analyze and add to discussion.

Line 193.- Not all FUBs are cited into the abstract cite them with specific results. Why only FUB1 and UB3? Explain at discussion.

Line 197.- Why 7 days post treatment? Dis you make previous kinetics? Add and explain.

Line 210.- cite Livak et al., method for -2 landa CT QRT-PCR calculates.

Add here number of replicates and statistical “n” used.

Statistical

Specify at each case ANOVA, T-Test, number “n” in all figure legends applied.

Results

Lines 235, 252,264…etc: please cite at all sections each figure separately.

Line 280, lines 294-295., move to discussion, try to not put conclusions into results section.

Section 334

Consider here to conclude that the most toxic metabolites for the fungus are also inducing defense mycotoxins to stop plant defense response, there is a clear correlaction between the induction of FUBs produced by the metabolites and their toxic effects on fungal growth. Please consider this point of view.

Line 337.- consider add those data to abstract.

Line 350.- move to abstract and discuss those results.

Section 356 to 367:

This section is just the more important of the article giving very interesting conclusison for me more than that you explained.

A pathview program analysis is required to look for correlations between the kinetic of plant response production and metabolism and fungal gene induction and cycle of infection.

Consider moving to discussion section except to data obtained.

Here the results described about coumarin probably indicate that coumarin has in plant a direct toxic effect on the fungus reducing the mycelial growth and biomass according to the natural function of coumarin on roots and this very high toxic effect on fungal growth makes that the fungus try to protect itself by producing higher amounts of mycotoxins. This is a combat between not well considered at this section. Similar effects of hydroxycoumarin are according to that and analyze if synergistic effects of both metabolites has been described previously.

Please consider this point of view making the results of this work much more relevant.

Discussion

Lines 370 to 376.- consider for changing the abstract.

Line 379-381.- Consider that metabolite production is very expensive for the plant, and it will try first to stop the fungus but if this low concentration not work they produce higher quantities, at low concentrations if the compound are very toxic the fungus will try to survive attaching the plant, consider this “combat” at the discussion.

Line 398.- Correlate metabolite production at plants with pathview transcriptional previous data and metabolomics data responding to Fusarium sp.

Line 406.- Which are the natural concentrations in plants? Check and explain and discuss here.

Line 407-408.- consider that the activity of metabolites will be lower along the time because molecular stability and interaction with PDA medium ??

Lines 413 to 415.- explain clearly, this is not well understood. Explain differences between total biomass and mycelial, most of biomass Is mycelia? Explain. Increasing of mycelial might indicate the high toxic effect of the chemical and activation of fungal metabolism to survive?

Line 421.- explain in detail stimulating process of biomass production.

Lines 431.- consider the toxic direct effect of coumarin more than the autotoxicity and correspondence with expressions found in FoSir5 gene??

Lines 440 to 455.- Consider the defense responses of the fungus to survive to plant defense production of toxic metabolites associated to infection progress. Contrast your results with previous transcriptional and metabolomic studies made againt Fol and Foz previously, consider the kinetics of fungal penetration and kinetic of plant production of those different metabolites analyzed, I consider that probably you will find very interesting results.

Prepare only ONE scheme simpler with the hypothesis to add it into principal article discussion section.

Line 445.- Hydroxycoumarin does not like to contribute to fungal infection try to get different conclusions according to plant defense and plant-fungal combat. Justify in terms of interaction at nature not on plates. Highlight that PDA is not the same than soil interactions and results should not be extrapolated.

Line 474.- consider the protection role of coumarin against the fungus also inhibiting FA synthesis and add to abstract. One of the more interesting results are the coumarin role in plant defense please highlight this into abstract or including into the title.

Lines 486 to 491.- consider that plant is trying to survive attaching the fungus.

Plant metabolites activate cytotoxic fungal responses and for me this is one of the most important results shown at this work and this should be highlighted at title, abstract and conclusions.

Line 494.- Consider that this is on plates not at nature but highlight the role of those metabolites in plant for protecting against Foz more than increasing the incidence and plant autotoxicity.

Lines 500 to 503.- I am not according to these conclusions please review and consider changing the conclusions and point of view about very interesting results obtained.

Hoping to be useful

Sincerely

Reviewer #2: I have carefully reviewed the manuscript entitled “In vitro Study of the Growth, Reproduction, and Pathogenicity Responses of Fusarium oxysporum f. sp. zingiberi to Autotoxins from Ginger.” This study is relevant and of potential interest, as it demonstrates that four ginger-related autotoxins differentially affect the growth, reproduction, and virulence-associated traits of Fusarium oxysporum, thereby contributing to Fusarium wilt under long-term ginger monoculture. The work further provides mechanistic insight into how these autotoxins may influence disease development. Overall, the manuscript is within the scope of the journal. However, the presentation and writing of the manuscript require substantial improvement, and several important concerns should be addressed before the manuscript can be considered for publication

6. PLOS authors have the option to publish the peer review history of their article (what does this mean?). If published, this will include your full peer review and any attached files.

Reviewer #1:**Yes:** Marta Berrocal-Lobo, Universidad Politécnica de Madrid

Reviewer #2: No

---

## [Author Response · Author response to Decision Letter 1]

1 May 2026

Our detailed, point-by-point responses to each comment have been submitted under the document titled “Response to Reviewers.” Please find the separated file in our submission.

---

## [Decision Letter · Decision Letter 1]

12 May 2026

Physiologic and molecular response of Fusarium oxysporum f. sp. zingiberi to ginger autotoxins

PONE-D-25-63719R1

Dear Dr. Naicheng,

We’re pleased to inform you that your manuscript has been judged scientifically suitable for publication and will be formally accepted for publication once it meets all outstanding technical requirements.

Kind regards,

Eugenio Llorens

Academic Editor

PLOS One

Additional Editor Comments (optional):

Reviewers' comments:

Reviewer's Responses to Questions

**Comments to the Author**

1. If the authors have adequately addressed your comments raised in a previous round of review and you feel that this manuscript is now acceptable for publication, you may indicate that here to bypass the “Comments to the Author” section, enter your conflict of interest statement in the “Confidential to Editor” section, and submit your "Accept" recommendation.

Reviewer #1: All comments have been addressed

Reviewer #2: All comments have been addressed

2. Is the manuscript technically sound, and do the data support the conclusions?

Reviewer #1: Yes

Reviewer #2: Yes

3. Has the statistical analysis been performed appropriately and rigorously? 

Reviewer #1: Yes

Reviewer #2: Yes

4. Have the authors made all data underlying the findings in their manuscript fully available?

Reviewer #1: Yes

Reviewer #2: Yes

5. Is the manuscript presented in an intelligible fashion and written in standard English?

Reviewer #1: Yes

Reviewer #2: Yes

6. Review Comments to the Author

Reviewer #1: Dear authors,

The article entitled “In vitro study of the growth, reproduction and pathogenicity responses of Fusarium oxysporum f. sp. zingiberi to autotoxins from ginger” has been corrected accordingly with results and right conclusions now fit with them. I consider that all my suggestions were considered.

Reviewer #2: (No Response)

7. PLOS authors have the option to publish the peer review history of their article (what does this mean?). If published, this will include your full peer review and any attached files.

Reviewer #1:**Yes:** Marta Berrocal-Lobo

Reviewer #2: No

---

## [Editor Report · Acceptance letter]

PONE-D-25-63719R1

PLOS One

Dear Dr. Li,

I'm pleased to inform you that your manuscript has been deemed suitable for publication in PLOS One. Congratulations! Your manuscript is now being handed over to our production team.

Kind regards,

on behalf of

Dr. Eugenio Llorens

Academic Editor

PLOS One